# LIA-X:

# INTERPRETABLE LATENT PORTRAIT ANIMATOR

## ABSTRACT

We introduce LIA-X, a novel interpretable portrait animator streamlined to transfer facial dynamics from a driving video to a source portrait, allowing for fine-grained control. LIA-X is an autoencoder that models motion transfer as a linear navigation of motion codes in the latent space. Crucially, it incorporates a novel Sparse Motion Dictionary that enables the model to disentangle facial dynamics into human interpretable motion. Deviating from previous 'warp-render' approaches, the interpretability of the Sparse Motion Dictionary allows LIA-X to support a highly controllable 'edit-warp-render' strategy, enabling precise manipulation of fine-grained facial semantics in the source portrait. This is instrumental in mitigating differences with the driving video *w.r.t.* pose and expression. In addition, we demonstrate the scalability of LIA-X by successfully training a large-scale model with approximately 1 billion parameters on extensive datasets. Experimental results suggest that our proposed method outperforms previous approaches in both self-reenact and cross-reenactment tasks across several benchmarks. The interpretable and controllable nature of LIA-X supports practical applications such as fine-grained, user-guided image and video editing, as well as 3D-aware portrait video manipulation. *We provide result videos on a visualization page* index.html, *included in the supplementary material*.

## 1 INTRODUCTION

The significant advancements in deep generative models (Ho et al., 2020; Song et al., 2021; Goodfellow et al., 2014) have led to remarkable progress in video generation. Portrait animation constitutes a domain-specific video generation task aimed at transferring facial dynamics from a driving video to a portrait image, receiving increased attention due to wide applications in entertainment, e-education, and digital human creation. In this context, towards accurately transferring facial dynamics, one strategy has been to leverage pre-computed explicit representations such as facial landmarks (Wang et al., 2019; Chang et al., 2023; Ma et al., 2024), 3DMM (Liu et al., 2019; Chen et al., 2021), optical flow (Li et al., 2018; Ohnishi et al., 2018) and dense poses (Xu et al., 2023) as motion guidance. Such a strategy has been widely utilized in both, GAN-based and recent diffusion-based methods (Chang et al., 2023; Ma et al., 2024). However, the generation quality of such methods is highly impacted by the performance of off-the-shelf feature extractors, strongly restricting usability in more challenging, real-world scenarios.

In addition, self-supervised learning-based techniques have been proposed to tackle this problem. Previous methods explored learning either explicit structures such as 2D/3D keypoints (Siarohin et al., 2019; 2021; Zhao & Zhang, 2022; Wang et al., 2021a; Guo et al., 2024a) or implicit motion codes (Wang et al., 2024; 2022) in an end-to-end manner, aimed at modeling facial dynamics. They typically follow a *'warp-render'* strategy that conducts portrait animation based on computed optical-flow fields. While these methods have achieved promising results in both, motion transferring and identity preservation, related performance drops significantly, in the case of large variations between source and driving data *w.r.t.* head pose and facial expression.

Motivated by the above, we introduce LIA-X, a novel framework that learns interpretable motion semantics, allowing for fully controllable *'edit-warp-render'* animation. Deviating from standard *'warp-render'* pipelines, LIA-X firstly leverages the learned motion representations to align source portrait with the initial driving frame, and then proceeds to transfer the final motion, see Fig. 1.

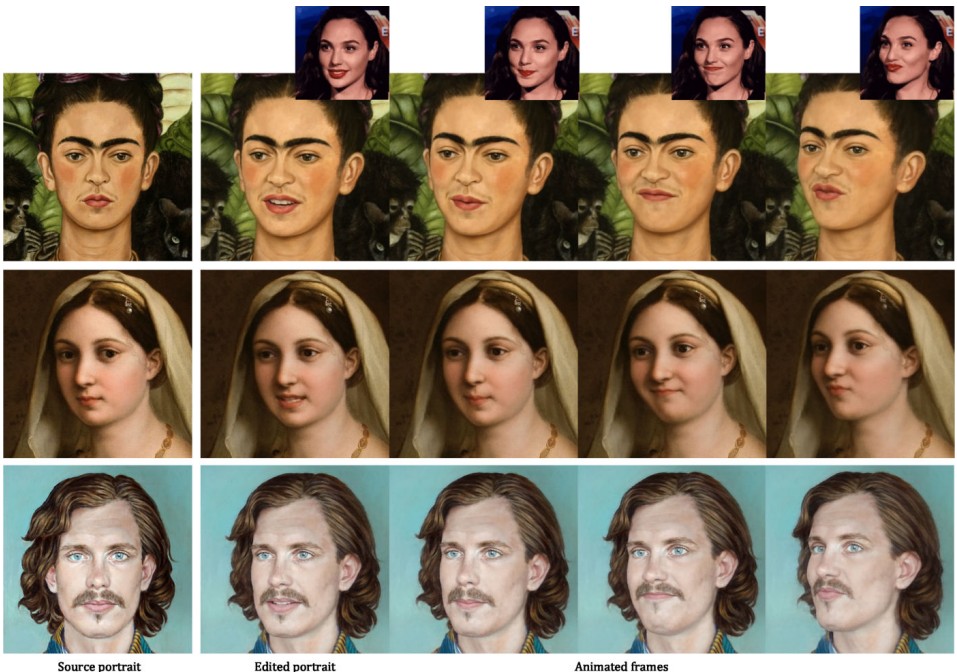

Figure 1: **Portrait Animation.** Results of three portraits animated by LIA-X using the video of Gal Gadot (small top right). Given a source portrait (1st column), LIA-X allows for editing the source portrait based on the learned semantics to align it with the initial driving frame *w.r.t.* head pose and facial expression (2nd column). The animated sequences (2nd-5th columns) are then obtained by applying motion transfer on the edited portraits.

Specifically, LIA-X is designed as a self-supervised autoencoder that does not necessitate explicit structure representations. Inspired by sparse dictionary coding (Olshausen & Field, 1996), we introduce a *novel Sparse Motion Dictionary* - a set of motion vectors in the autoencoder that capture the underlying motion distribution. The sparsity constraint encourages the model to utilize a minimal set of motion vectors, in order to reconstruct the training images, endowing the motion vectors with *enhanced interpretability* as opposed to dense motion dictionaries (Wang et al., 2024; 2022). The animation process is then modeled as a linear navigation of these motion codes, in order to obtain optical-flow fields for warping the source portrait. Notably, our proposed interpretable motion vectors can be directly leveraged at the inference stage towards manipulating source portraits, supporting fine-grained edits of facial attributes (*e.g.,* eyes and mouth), as well as 3D-aware transformation (*e.g.,* yaw, pitch and roll) in both, image and video domains.

*LIA-X is the first model to self-learn implicit semantic representations specifically tailored for controllable portrait animation and editing, to the best of our knowledge.*

Further, we analyze the scalability of proposed LIA-X framework and successfully train large-scale models with up to 1 billion parameters using a diverse mixture of public and internal talking head datasets. Experimental results demonstrate that scaling up LIA-X significantly improves performance across several benchmarks. Given faster inference speed of autoencoder-based models compared to diffusion-based approaches, we advocate that the scalable and interpretable design of LIA-X can serve as a valuable complement to current state-of-the-art generative models, enabling efficient and highly controllable video generation. In summary, key contributions of this work include the following.

- We propose a novel portrait animator, LIA-X, an autoencoder that incorporates an interpretable Sparse Motion Dictionary to enable controllable portrait animation via an *'edit-warp-render'* strategy.
- We analyze the scalability of LIA-X and demonstrate that our design can be scaled up to achieve superior performance.

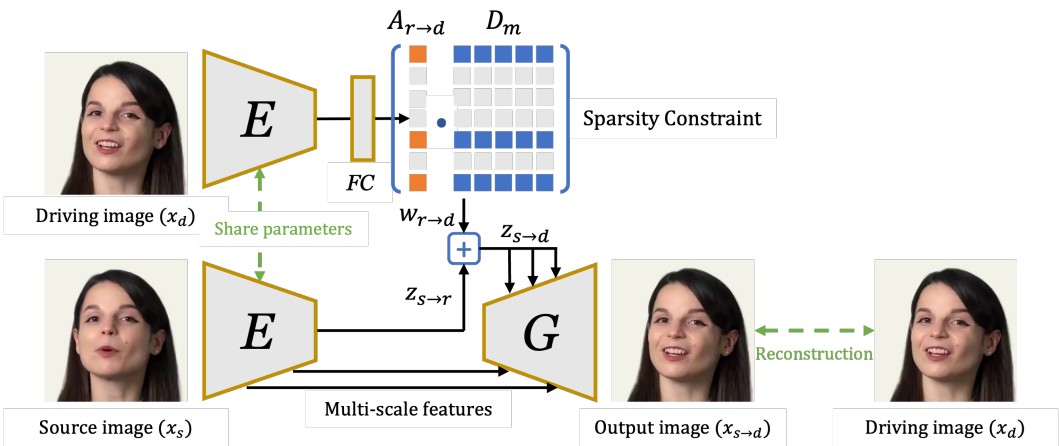

Figure 2: **Overview of LIA-X.** LIA-X incorporates an encoder $E$, a generator $G$ that includes an optical-flow generator $G_f$ and a rendering network $G_r$. LIA-X is trained using a self-supervised learning strategy. Towards obtaining an interpretable motion dictionary $D_m$, a sparsity constraint is included into the training objective. This encourages the network to utilize a minimal set of vectors in $D_m$ to reconstruct each driving image.

- Extensive experiments show that LIA-X outperforms state-of-the-art methods across multiple datasets, while also enabling a range of applications, including image and video editing, as well as 3D-aware portrait manipulation.

## 2 PRELIMINARY

Latent Image Animator (LIA) (Wang et al., 2022; 2024) was designed as an autoencoder consisting of an encoder $E$, an optical-flow generator $G_f$ and a rendering network $G_r$ aiming to transfer motion of a talking head to a still portrait via self-supervised learning. LIA models motion as linear navigation of motion codes in a latent space, and follows the '*warp-render*' strategy to generate optical-flow fields via $G_f$ and render the animated result via $G_r$.

**Linear Navigation.** LIA models motion transferring as learning transformations from source to driving image $x_s \rightarrow x_d$. It proved that for any given image, there exists an 'implicit reference image' $x_r$, and the transformation can be modeled as $x_s \rightarrow x_r \rightarrow x_d$ in an implicit manner. The transformation is modeled as motion code $z_{s \rightarrow d}$ in latent space, and with the help of the reference space, it can be represented as a linear navigation from $z_{s \rightarrow r}$ along a path $w_{r \rightarrow d}$, denoted as

$$z_{s \rightarrow d} = z_{s \rightarrow r} + w_{r \rightarrow d}, \tag{1}$$

where $z_{s \rightarrow r}$ indicates the transformation from source image to reference image, and is obtained via $E(x_s) = z_{s \rightarrow r}$. To learn $w_{r \rightarrow d}$, a learnable motion dictionary $D_m$ consisting of a set of orthogonal motion vectors $\{\mathbf{d_1}, ..., \mathbf{d_M}\}$ was proposed. Any linear path in the latent space can be represented as

$$w_{r \rightarrow d} = \sum_{i=1}^{M} a_i \mathbf{d_i}, \tag{2}$$

where $\mathcal{A}_{r \rightarrow d} = \{a_1, ..., a_M\}$ indicates the magnitude for each motion vector, and is obtained via $\mathcal{FC}(E(x_d)) = \mathcal{A}_{r \rightarrow d}$. The linear navigation from any $x_s$ to $x_d$ can be represented as

$$z_{s \rightarrow d} = z_{s \rightarrow r} + \sum_{i=1}^{M} a_i \mathbf{d_i}. \tag{3}$$

**Image Animation.** Once $z_{s \rightarrow d}$ is obtained, optical flow is generated via $G_f(z_{s \rightarrow d}) = \phi_{s \rightarrow d}$. The source image $x_s$ is warped based on $\phi$, and the warped features are rendered through $G_r$ to produce the generated image

$$x_{s \rightarrow d} = G_r(\mathcal{T}(\phi_{s \rightarrow d}, x_s)), \tag{4}$$

where $\mathcal{T}$ indicates the warping operation. In practice, multi-scale optical flow is generated to warp multi-scale feature maps of $x_s$.

**Learning.** The objective of self-supervised learning consists of three parts, an $L_1$ reconstruction loss, a VGG-based perceptual loss and an adversarial loss between $x_{s \to d}$ and $x_d$

$$\mathcal{L}(x_{s \to d}, x_d) = \mathcal{L}_{recon}(x_{s \to d}, x_d) + \lambda \mathcal{L}_{vgg}(x_{s \to d}, x_d) \\ + \mathcal{L}_{adv}(x_{s \to d}). \tag{5}$$

**Inference.** LIA proposed a unified formulation for self-reenactment and cross-reenactment at the inference stage

$$z_{s \to t} = (z_{s \to r} + w_{r \to s}) + (w_{r \to t} - w_{r \to 1}), \ t \in \{1, ..., T\}. \tag{6}$$

For self-reenactment, where $w_{r \to s} = w_{r \to 1}$, Eq. 6 can be simplified as

$$z_{s \to t} = z_{s \to r} + w_{r \to t}, \ t \in \{1, ..., T\}, \tag{7}$$

which is the same as the training stage.

For cross-reenactment, where $w_{r \to s} \neq w_{r \to 1}$, Eq. 6 can be reformulated as

$$z_{s \to t} = \underbrace{z_{s \to s}}_{\text{reconstruction}} + \underbrace{(w_{r \to t} - w_{r \to 1})}_{\text{motion difference}}, \ t \in \{1, ..., T\}, \tag{8}$$

where the animation process is modeled as *linearly navigating the source image along the driving motion direction in the latent space*.

## 3 METHODOLOGY OF LIA-X

We proceed to introduce our proposed LIA-X, including the general model architecture, the Sparse Motion Dictionary, as well as the interpretable and controllable animation capabilities.

### 3.1 ARCHITECTURE

LIA-X follows the general architecture design of LIA, containing an encoder $E$, an optical-flow generator $G_f$, and a rendering network $G_r$. However, deviating from previous methods (Siarohin et al., 2019; 2021; Zhao & Zhang, 2022) that rely on simple and small-scale networks, we design the LIA-X architecture to be more scalable by incorporating advanced residual blocks inspired by StyleGAN-T (Sauer et al., 2023) in both, encoder and generators.

### 3.2 SPARSE MOTION DICTIONARY

While the original motion dictionary in LIA contains certain semantic meanings, we observed that it is difficult to disentangle and independently control individual factors such as mouth movements, eyebrow dynamics, etc. Multiple semantic motions are entangled and challenging to leverage for controllable image and video editing tasks.

Drawing from *sparse dictionary coding*, we propose a Sparse Motion Dictionary aimed at improving the interpretability of motion representations. Our findings indicate that this simple enhancement greatly improves the interpretability of the motion vectors, which facilitates more precise control over animation.

Specifically, we introduce a sparse penalty $S(\cdot)$ on the motion coefficients $\mathcal{A}_{r \to d}$ that encourages the model to employ a minimal number of motion vectors to reconstruct the images during self-supervised training. The overall learning objective of LIA-X is then

$$\mathcal{L}(x_{s \to d}, x_d) = \mathcal{L}_{recon}(x_{s \to d}, x_d) + \lambda_1 \mathcal{L}_{vgg}(x_{s \to d}, x_d) \\ + \mathcal{L}_{adv}(x_{s \to d}) + \lambda_2 S(\mathcal{A}_{r \to d}), \tag{9}$$

where $\lambda_1$ and $\lambda_2$ denote coefficients to balance the losses, and we implement $S(\cdot)$ as the $L_1$ norm.

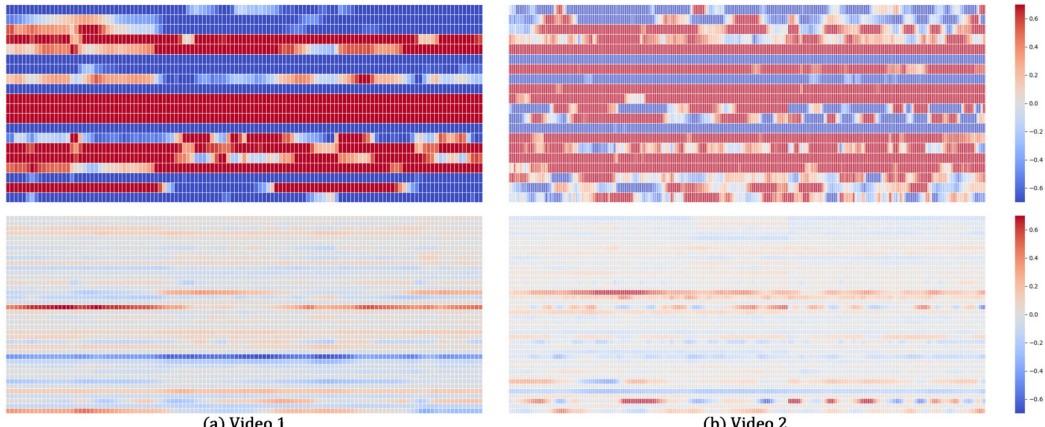

(a) Video 1         (b) Video 2

Figure 3: **Sparsity Analysis.** We show a comparison of the $\mathcal{A}_{r \to s}$ activations of two videos between two models - one trained without sparse motion dictionary (top), and the other with a sparse motion dictionary (down). It can be clearly observed that when learned without a sparsity constraint, the model reconstructs each frame by activating nearly all motion vectors. In contrast, the model trained with a sparsity constraint selects only a few vectors to be activated for each reconstruction.

### 3.3 CONTROLLABLE INFERENCE

While Eq. 8 enables successful motion transfer from the driving video to the source portrait, achieving high-quality generation necessitates a critical condition: *source and initial driving frames must exhibit similar head poses and facial expressions*. However, in a number of real-world applications, this requirement cannot be satisfied. With the interpretable Sparse Motion Dictionary, LIA-X is able to address this issue by using an *"edit-warp-render"* strategy.

Consequently, we first edit the source portrait using the corresponding motion vectors, ensuring that the head pose and facial expression closely match those of the driving image. Then, the animation process can be formulated as follows.

$$z_{s \to t} = \underbrace{z_{s \to \mathcal{E}(s)}}_{\text{editing}} + \underbrace{(w_{r \to t} - w_{r \to 1})}_{\text{motion difference}}, \ t \in \{1, ..., T\}, \tag{10}$$

where $\mathcal{E}(\cdot)$ indicates the editing operation on the source image. This allows LIA-X to better handle large variations between source and driving data, leading to higher-quality and more controllable portrait animation.

## 4 EXPERIMENTS

In this section, we proceed to present a qualitative analysis on sparsity, interpretability, and controllability of the proposed LIA-X framework. In addition, we present comparative results between LIA-X and state-of-the-art methods such as FOMM, TPS, DaGAN, LIA, MCNet, X-Portrait, and LivePortrait, in the setting of portrait animation. Finally, we quantitatively evaluate LIA-X and compare it with prior work on two important tasks, *self-reenactment* and *cross-reenactment*. Details of implementation and training datasets are presented in Appx. A.2 and A.3.

### 4.1 SPARSITY ANALYSIS

To showcase the effectiveness of our proposed Sparse Motion Dictionary, we visualize motion coefficient vectors $\mathcal{A}_{r \to s}$ for two videos in Fig. 3. We compare the results of models with and without sparsity constraint in the motion dictionary.

We clearly observe that the model without sparse motion dictionary activates nearly all motion vectors, indicating a lack of selectivity for different input data. The semantics are entangled within each motion vector. In contrast, for the model with Sparse Motion Dictionary, sparsity can be clearly

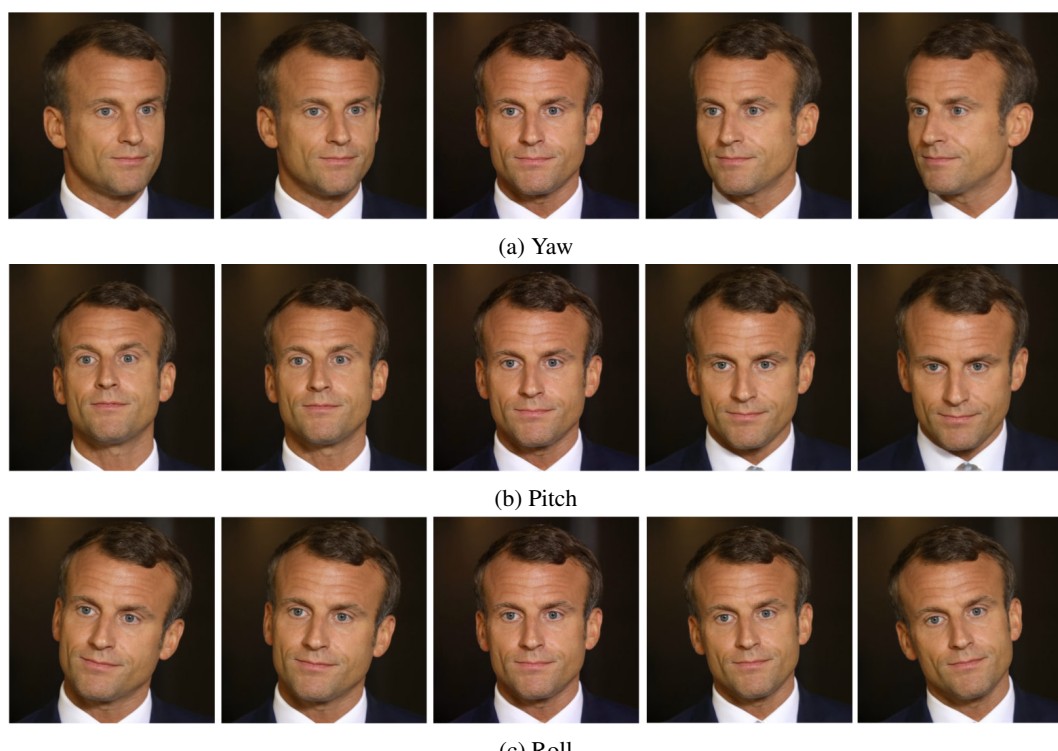

(a) Yaw

(b) Pitch

(c) Roll

Figure 4: **3D-aware Portrait Manipulation.** We illustrate 3D-aware manipulation capabilities for a single identity. By manipulating corresponding motion vectors, LIA-X can successfully perform (a) *yaw*, (b) *pitch* and (c) *roll* modifications, without necessitating additional 3D representations.

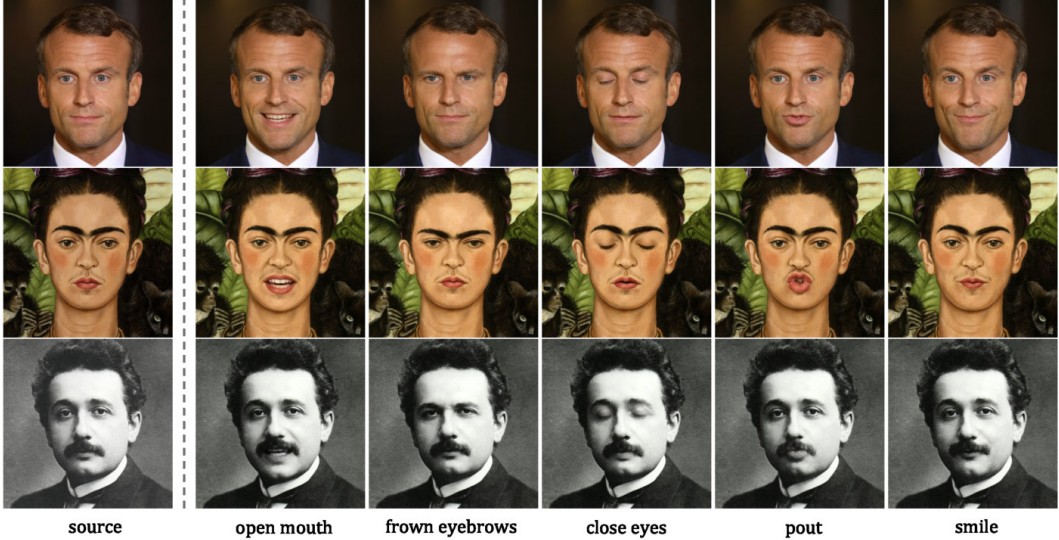

| source | open mouth | frown eyebrows | close eyes | pout | smile |

Figure 5: **Image Editing.** Fine-grained semantic attributes, such as *open/close mouth*, *frown/raised eyebrows*, *open/close eyes*, *pout*, and *smile* can be effectively controlled by manipulating the corresponding motion vectors.

observed in $\mathcal{A}_{r \to s}$, as only few vectors are active, whereas the contributions of others can be largely omitted. These results clearly prove the effectiveness of the constraint in improving the sparsity of the motion dictionary.

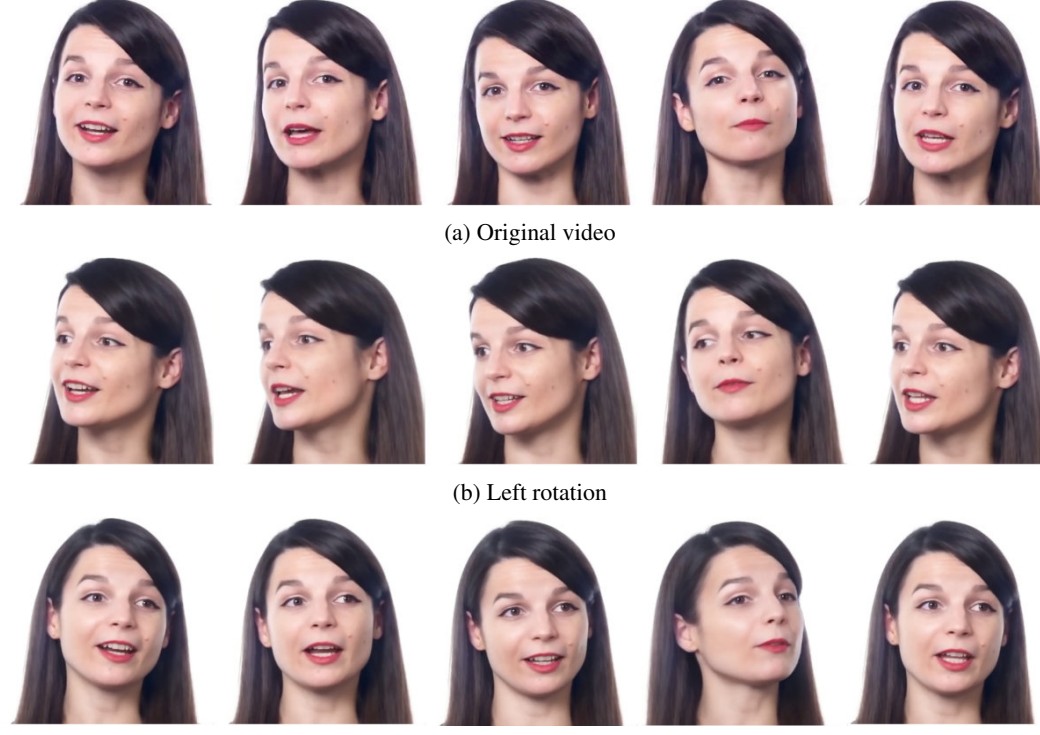

(a) Original video

(b) Left rotation

(c) Right Rotation

Figure 6: **3D-aware Video Manipulation.** We demonstrate the use of learned 3D-aware semantics to manipulate a real-world video. The original video (a) has been successfully rotated to left (b) and right (c) respectively, without using any explicit 3D representation, and the identity of the original video subject has also been well-preserved throughout these manipulations.

## 4.2 INTERPRETABILITY AND CONTROLLABILITY ANALYSIS

While we have proven the ability to obtain a Sparse Motion Dictionary, it is crucial to understand whether each vector in the dictionary is interpretable and controllable. To investigate the semantic meanings of the motion vectors, we linearly manipulate the vector $d_i$ by

$$z_{s \to \mathcal{E}(s)} = z_{s \to s} + a_i \mathbf{d_i}, \tag{11}$$

where we set $a_i$ as a small perturbation ranging from -0.5 to 0.5 with a step of 0.1. Surprisingly, we find that the semantics are well-disentangled in the motion dictionary and can be easily manipulated. Almost all activated vectors correspond to human-understandable motions. Fig. 4 illustrates examples of LIA-X performing 3D-aware manipulation such as *yaw*, *pitch* and *roll*, without relying on any explicit 3D representation during training or inference. We provide a comparison between LIA and LIA-X in motion dictionary manipulation in App. A.4.

**Image Editing.** We further utilize Eq. 11 to manipulate more motion vectors and demonstrate the results in Fig. 5. Besides 3D-aware semantics, LIA-X can control various fine-grained attributes such as 'open/close mouth', 'frown/raise eyebrows', 'open/close eyes', 'pout', 'smile', etc. Such interpretable vectors are automatically disentangled through self-supervised learning. With such a powerful editing capability, LIA-X can be readily used as a portrait editing tool, allowing for linear combination of different semantic operations, allowing for complex editing tasks.

**Video Editing.** Given the ability to edit a single portrait, extending the editing to the video level becomes feasible. Given a real-world talking head video, we first apply Eq. 11 on the initial frame to manipulate the target semantics. Then, we use Eq. 10 to transfer the motion to the edited first frame. Fig. 6 demonstrates the results of using LICA-X to rotate the portrait in a video, where the identity is well-preserved while the head pose is seamlessly changed.

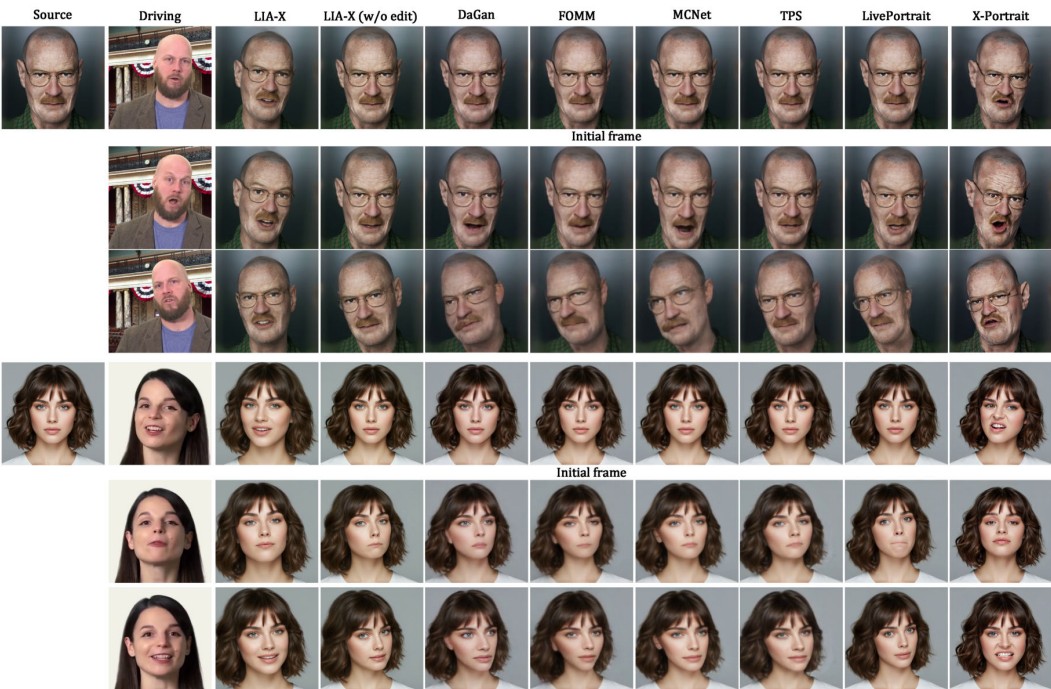

Figure 7: **Qualitative Comparison on Cross-reenactment.** The ability of LIA-X to edit the source portrait using interpretable motion vectors before animation allows it to correct initial misalignments with the driving frame. As shown here, this editing capability enables LIA-X to significantly outperform other approaches, particularly in cases with large variations of head pose and facial expression between source and driving data.

| Method | VoxCelebHQ | | | | | TalkingHead-1KH | | | | |
|---|---|---|---|---|---|---|---|---|---|---|
| | L1 ↓ | LPIPS ↓ | SSIM ↑ | PSNR ↑ | FID ↓ | L1 ↓ | LPIPS ↓ | SSIM ↑ | PSNR ↑ | FID ↓ |
| | | | | | $256 \times 256$ resolution | | | | | |
| FOMM | 0.046 | 0.27 | 0.66 | 22.40 | 12.67 | 0.040 | 0.100 | 0.72 | 23.31 | 30.39 |
| DaGAN | 0.044 | 0.110 | 0.69 | 23.04 | **9.13** | 0.036 | 0.088 | 0.77 | 24.95 | 25.50 |
| TPS | 0.043 | 0.112 | 0.70 | 23.24 | 10.82 | 0.037 | 0.089 | 0.77 | 24.56 | 28.05 |
| MCNet | 0.040 | 0.176 | 0.72 | 23.73 | 18.63 | **0.030** | 0.097 | **0.79** | 25.70 | 28.06 |
| LIA-X | **0.036** | **0.095** | **0.73** | **24.82** | 10.74 | 0.035 | **0.086** | 0.78 | **26.26** | **24.71** |
| | | | | | $512 \times 512$ resolution | | | | | |
| X-Portrait | 0.110 | 0.302 | 0.56 | 16.99 | 19.92 | 0.058 | 0.134 | 0.63 | 19.46 | 41.19 |
| LivePortrait | 0.087 | 0.264 | 0.67 | 17.45 | 12.90 | 0.052 | 0.120 | 0.73 | 20.26 | 39.98 |
| LIA | 0.052 | 0.211 | 0.68 | 22.14 | 21.86 | 0.049 | 0.165 | 0.72 | 23.37 | 44.64 |
| LIA-X | **0.040** | **0.160** | **0.75** | **24.39** | 12.50 | **0.035** | **0.115** | **0.80** | **26.07** | 38.93 |

Table 1: **Quantitative Evaluation for Self-Reenactment.** We compare the performance of LIA-X against state-of-the-art methods on two different resolutions across two datasets.

**Portrait Animation.** We qualitatively compare LIA-X with state-of-the-art methods on cross-identity portrait animation, see Fig. 7. Comparison between LIA-X and LIA is portrayed in Fig. A.4. Visualizations showcase the effectiveness of the editing stage of LIA-X, which can significantly improve generated results by aligning source portrait with the initial driving frame. This editing capability allows for a more controllable animation process as opposed to previous methods. In contrast, other state-of-the-art methods struggle in case that source and driving data encompass large variations in head pose, facial expression, as well as identity. The performance of such methods degrades dramatically in such scenarios, whereas LIA-X maintains high-quality and controllable animation results by leveraging its interpretable motion representations and the tailored "edit-warp-render" pipeline.

| Method | ID Similarity ↓ | Image Quality ↑ |
|---|---|---|
| FOMM | 0.262 | 37.08 |
| DaGAN | 0.272 | 39.30 |
| TPS | 0.216 | 38.27 |
| MCNet | 0.252 | 37.88 |
| X-Portrait | 0.217 | 55.41 |
| LivePortrait | 0.243 | 51.41 |
| LIA-X | **0.206** | **58.74** |

Table 2: **Quantitative Evaluation for Cross-Reenactment.** We compare the performance of LIA-X with state-of-the-art methods on the constructed dataset for cross-reenactment scenarios.

| Model | L1 ↓ | LPIPS ↓ | SSIM ↑ | PSNR ↑ |
|---|---|---|---|---|
| TalkingHead-1KH | | | | |
| Base (0.05B) | 0.042 | 0.13 | 0.77 | 24.98 |
| Middle (0.3B) | **0.035** | **0.113** | 0.79 | 25.84 |
| Large (0.9B) | **0.035** | 0.115 | **0.80** | **26.07** |
| VoxCelebHQ | | | | |
| Base (0.05B) | 0.043 | 0.171 | 0.72 | 23.62 |
| Middle (0.3B) | **0.040** | **0.16** | 0.74 | 24.31 |
| Large (0.9B) | **0.040** | **0.16** | **0.75** | **24.39** |

Table 3: **Scalability analysis** on TalkingHead-1KH and VoxCelebHQ. We train three variations of LIA-X with 0.05B, 0.3B, and 0.9B parameters, respectively.

### 4.3 QUANTITATIVE EVALUATION

We quantitatively compare LIA-X with state-of-the-art (SOTA) methods on two tasks, *self-reenactment* and *cross-reenactment*. To ensure a fair comparison, we train LIA-X at two different resolutions, $256 \times 256$ and $512 \times 512$, and compare with the corresponding methods.

**Self-Reenactment.** We evaluate our method on the validation sets of VoxCelebHQ (Wang et al., 2024) and TalkingHead-1KH (Wang et al., 2021a), which contain 483 and 25 videos respectively. We reconstruct each video sequence utilizing the first frame as the source image and the entire video as the driving data. We report results using several metrics, L1, LPIPS, SSIM, PSNR, and FID. As shown in Tab. 1, on both high and low resolutions, LIA-X outperforms both GAN-based and diffusion-based methods across all metrics.

**Cross-Reenactment.** To construct the validation set for cross-reenactment, we select 70 videos from the HDTF dataset as driving data, and per video, we randomly select 2 images from the AAHQ dataset (Liu et al., 2021) as the source data, resulting in a total of 140 videos. Since there is no ground truth data for this task, we evaluate the results using two metrics, *Identity Similarity* and *Image Quality*. While Identity Similarity indicates the average embedding difference between each generated frame and source portrait, Image Quality is computed following (Su et al., 2020) to indicate the generated image quality. As shown in Tab. 2, LIA-X outperforms other methods in both metrics, proving its ability to naturally transfer motion, while preserving the original identities.

### 4.4 ABLATION STUDY ON SCALABILITY

Towards verifying the effectiveness of scalability, we conduct an ablation study on the model size. We train three variations of LIA-X with 0.05B, 0.3B, and 0.9B parameters, respectively, maintaining the same training configuration. The three models follow the same architectural design and only differ in the number of residual blocks, channel numbers, and motion dictionary size. The results in Tab. 3 demonstrate that scaling is effective for improving model performance. Qualitative results can be found in App. A.5. However, we note that the improvement from 0.3 billion to 0.9 billion parameters becomes relatively minor. We hypothesize that this may be attributed to the current dataset size being insufficient to fully leverage the training of such large-scale model. To further enhance LIA-X's performance, larger and more diverse training datasets are essential.

## 5 CONCLUSIONS

In this work, we have introduced LIA-X, a novel portrait animator that incorporates an interpretable Sparse Motion Dictionary. This enables LIA-X to support a highly controllable *'edit-warp-render'* animation strategy. Furthermore, we have analyzed the scalability of the architecture of LIA-X, demonstrating its ability to achieve outstanding results when scaling up to larger model sizes, up to 1 billion parameters. Extensive evaluations show that LIA-X outperforms SOTA methods across several benchmarks, while also supporting diverse applications such as fine-grained image/video editing and high-quality portrait animation. We envision our technique to serve as a valuable complement to current generative approaches and provide a novel way for interpretable video generation.

## ETHIC STATEMENT

In this work, we aim to animate portraits using driving videos. Our designed LIA-X is an interpretable and controllable image animator. Our approach can be used for digital human, online education, and data synthesis for other computer vision tasks. We note that our framework mainly focuses on learning how to model motion distribution in an image animator rather than directly model appearance. Therefore, our framework is not biased towards any specific gender, race, region, or social class. It works equally well irrespective of the difference in subjects.

## REPRODUCIBILITY STATEMENT

We assure that all results shown in the paper and supplemental materials can be reproduced. We intend to open-source our code, as well as trained models.

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

# A APPENDIX

In this Appendix, we firstly discuss related work and outline the novelty in this work in App. A.1. Then, we describe implementation and details pertaining the training datasets in App. A.2 and A.3. Next, we compare LIA-X and LIA in linear manipulation of the motion dictionary, as well as qualitative results of portrait animation in App. A.4. We also qualitatively demonstrate the effectiveness of scaling LIA-X to larger model size in App A.5. Finally, we discuss about limitations in current model design and provide potential future directions in App A.7.

## A.1 RELATED WORK

Portrait animation has seen significant advancements in recent years, driven by the remarkable progress in deep generative models for video generation (Vondrick et al., 2016; Saito et al., 2017; Tulyakov et al., 2018; Wang et al., 2020c;b; Wang, 2021; Wang et al., 2021b; Clark et al., 2019; Brooks et al., 2022; Yu et al., 2022; Skorokhodov et al., 2022; Tian et al., 2021; Wang et al., 2023b; Chen et al., 2023; 2024; Guo et al., 2024b; Singer et al., 2023; Ho et al., 2022; Ma et al., 2025; Wang et al., 2023a; Blattmann et al., 2023; Menapace et al., 2024; Yan et al., 2021; Brooks et al., 2024; Zhang et al., 2024). Previous methods have explored learning structure representations either based on conditional generation approaches (Chan et al., 2019; Wang et al., 2018; Zakharov et al., 2019; Wang et al., 2019; Yang et al., 2020; Zhao & Zhang, 2022) relying on off-the-shelf extractors, or self-supervised learning strategies to learn representations such as 2D/3D keypoints (Siarohin et al., 2019; Wang et al., 2021a; Guo et al., 2024a; Zhao & Zhang, 2022), motion regions (Siarohin et al., 2021), and depth maps (Hong et al., 2022) in an end-to-end manner. Additionally, self-attention mechanism (Yuan et al., 2023) has been studied to improve cross-identity generation quality.

More recent techniques (Wang et al., 2024; 2022) have proposed to model motion transfer as a linear navigation of learned motion codes, which has proven effective in implicitly capturing 2D and 3D representations to capture complex facial dynamics. Diffusion-based models (Xie et al., 2024) have demonstrated efficacy in portrait animation, attributed to their robust generalization capabilities derived from pretraining on large-scale datasets.

However, one key limitation remains across all existing methods associated to the lack of an effective mechanism to align source portrait with initial driving frame *w.r.t.* head pose and facial expression. This renders their performance drop dramatically in case of large variations between source and driving data, further limiting their usability in more general real-world scenarios. Deviating from previous approaches, LIA-X incorporates an interpretable motion dictionary, which allows for the source portrait to be edited before animation in a controllable manner. With this unique capacity, LIA-X is instrumental in a number of tasks such as portrait animation, image and video editing, and 3D-aware portrait manipulation.

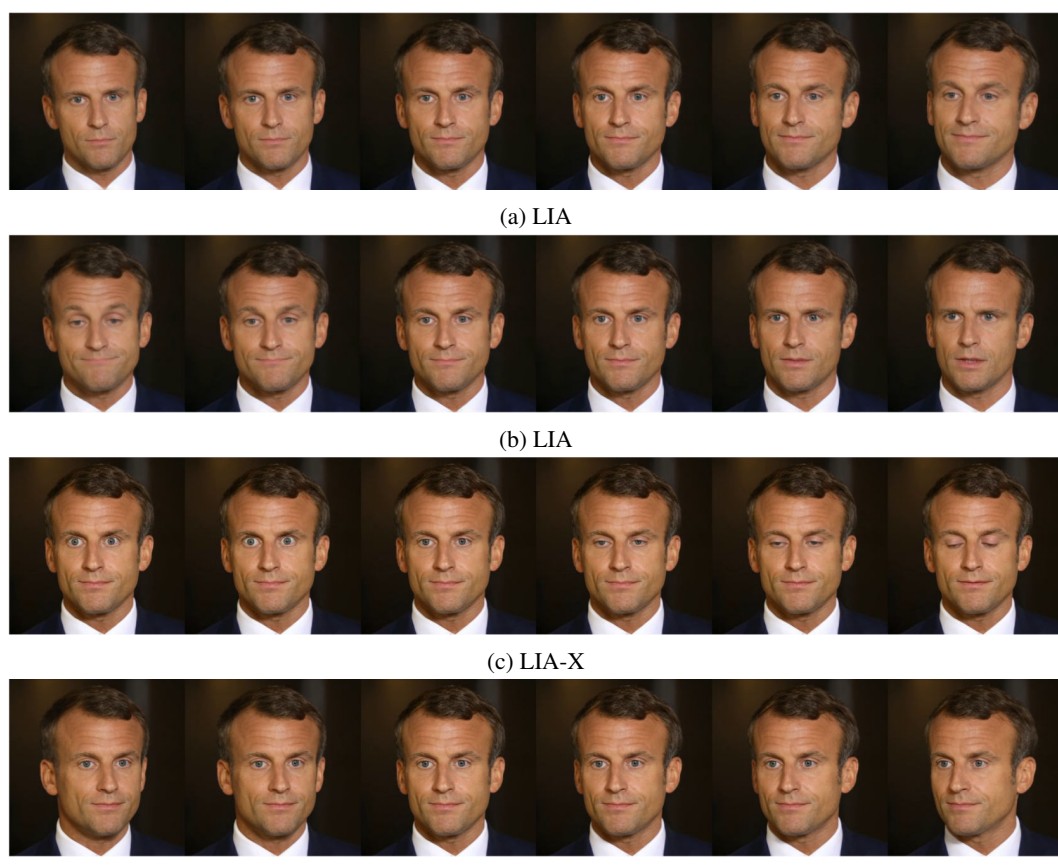

(a) LIA

(b) LIA

(c) LIA-X

(D) LIA-X

Figure 8: **Manipulation results between LIA and LIA-X.** We show linear manipulation of two vectors in both LIA and LIA-X, respectively. We observe that semantics in LIA's motion dictionary are entangled (in (a) and (b), head, eye and mouth are all altered in case that a single vector is being manipulated), whereas LIA-X is able to disentangle various semantics very well, for example open/close eyes (c) and yaw (d).

## A.2 IMPLEMENTATION DETAILS

We build LIA-X upon the architecture of LIA (Wang et al., 2024; 2022). To scale the model to larger sizes and prevent training instability, we design novel residual blocks for both, optical flow generator $G_f$, as well as rendering network $G_r$. Our scaling strategy focuses on increasing the number of channels, the depth of residual blocks, as well as using a larger motion dictionary compared to the original LIA. With these architectural enhancements, the largest model size of LIA-X reaches around 1 billion parameters. We train the entire LIA-X model using 8 A100 GPUs, and apply gradient accumulation to increase the effective batch size when training the larger-scale LIA-X variants.

## A.3 DATASETS

To scale the training data, we mix 4 publicly available datasets, namely VoxCelebHQ (Wang et al., 2024), TalkingHead-1KH (Wang et al., 2021a), HDTF (Zhang et al., 2021) and MEAD (Wang et al., 2020a). Additionally, we include one internally collected dataset. In total, our training dataset contains 0.5 million talking head sequences, comprising around 94 million frames and 55,000 different identities. Experiments show that this dataset scaling strategy enables LIA-X to achieve outstanding performance in generalizing to unseen portrait images.

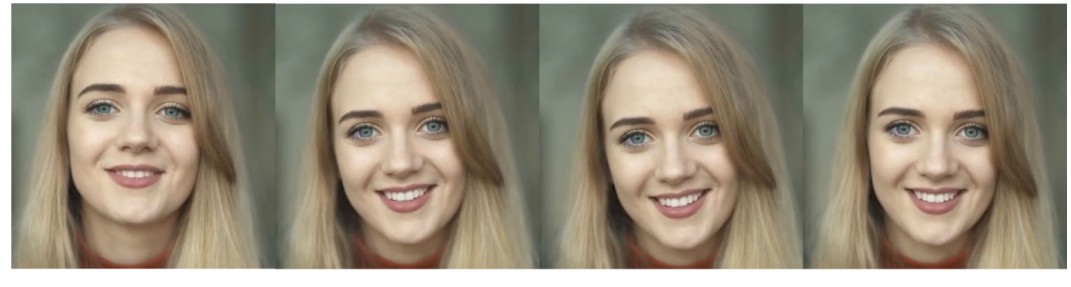

(a) Driving video

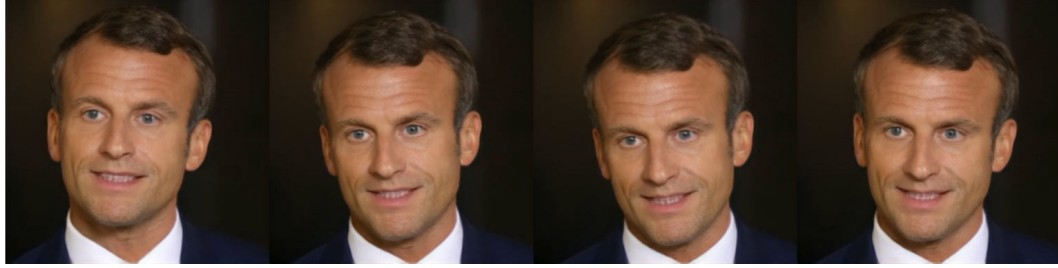

(b) LIA

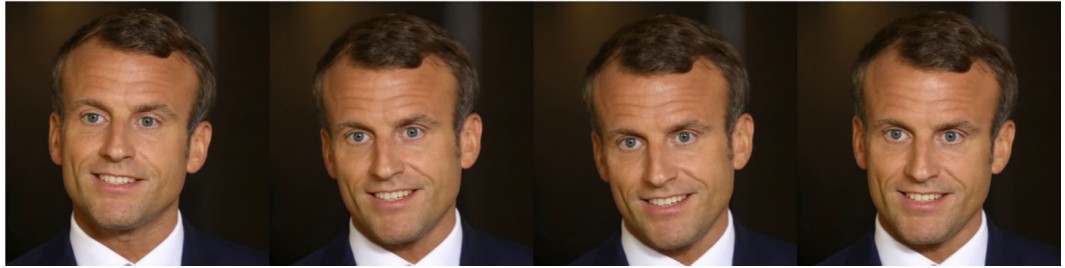

(c) LIA-X

Figure 9: **Comparison between LIA and LIA-X.** We compare the performance of LIA and LIA-X based on the same driving video and source image. LIA-X achieves better video quality than LIA.

## A.4 COMPARISON BETWEEN LIA AND LIA-X

We firstly show linear manipulation of vectors in motion dictionary in both, LIA and LIA-X. Fig. 8 demonstrates that semantics in LIA's motion vectors are entangled, as linear manipulation of a single vector affects eye, mouth and head. On the other side, we observe that semantics are well-disentangled, as manipulation of a single vector only affects one semantic (e.g., open/close eyes, yaw), which proves the effectiveness of our proposed sparse motion dictionary.

We also qualitatively show comparison between LIA and LIA-X on image animation in Fig. 9. By using the same driving video, LIA-X has achieved better performance than LIA w.r.t. video quality.

## A.5 SCALABILITY ANALYSIS

We qualitatively show animation results of LIA-X with different model size. Fig. 10 demonstrates that scaling LIA-X from 0.05B to 0.9B significantly improves generation quality, in particular *w.r.t.* details such as improved teeth, eye-balls and facial expression.

## A.6 INFERENCE SPEED AND MODEL SIZE

We make a comparison among state-of-the-art methods, as well as LIA-X on model size and inference speed on A6000 GPU in Tab 4. It can be observed that diffusion-based method (X-Portrait) is much heavier than other methods and has the lowest inference speed. LIA-X achieves a nice balance between inference speed and model size. Compared to LivePortrait, LIA-X (large) is 8 times larger

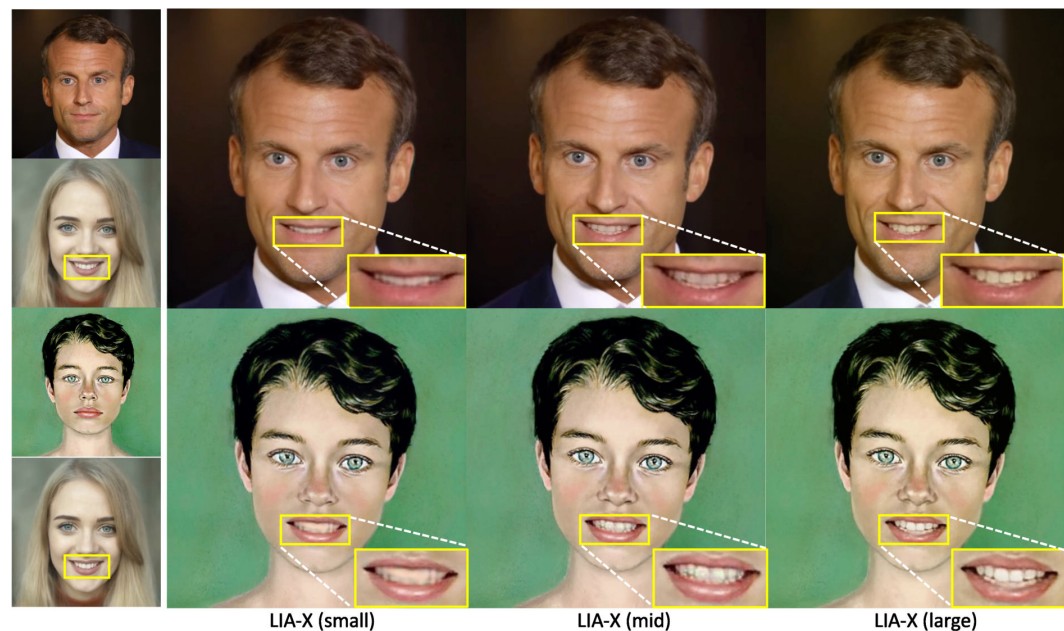

Figure 10: **Scalability Analysis.** Scaling is effective for generating details such as teeth and eyeballs. The large model (0.9B) outperforms the smaller model (0.05B).

| Method | FOMM | DaGAN | TPS | MCNet | X-Portrait | LivePortrait | LIA | LIA-X (small) | LIA-X (mid) | LIA-X (large) |
|---|---|---|---|---|---|---|---|---|---|---|
| Size (B) | 0.05 | 0.06 | 0.07 | 0.10 | 3.07 | 0.13 | 0.05 | 0.05 | 0.34 | 0.91 |
| Speed (s) | 0.014 | 0.027 | 0.026 | 0.035 | 4.700 | 0.114 | 0.030 | 0.151 | 0.159 | 0.23 |

Table 4: **Model size and inference speed.** We show model size an inference speed of current SOTA on portrait animation. The speed indicates generation time per frame.

on model size while the inference speed is only 2 times slower, which indicates the design of LIA-X is more efficient.

## A.7 LIMITATIONS AND FUTURE WORK

While our proposed method has achieved promising performance, there remain limitations that need to be tackled in future work. Firstly, as the current model can only be used on a fixed resolution, designing novel techniques to allow dynamic resolution may further improve performance. Secondly, our model follows the convolutional network architecture which may have limitations to further scale up. DiT-based (Peebles & Xie, 2022) architecture will be explored in future work for scalability.

## A.8 USE OF LARGE LANGUAGE MODELS

We clarify the involvement of large language models (LLMs) is only for improving and polishing the manuscript.

