# OpenReview forum: "LIA-X: Interpretable Latent Portrait Animator"
_ICLR.cc/2026/Conference — ICLR 2026 Conference Withdrawn Submission_

### Official Review · Reviewer_eZRG · 2025-10-28

**Soundness:** 3
**Presentation:** 3
**Contribution:** 3
**Rating:** 4
**Confidence:** 4

**Summary:**

This paper targets the task of portrait animation, proposing a GAN-based autoencoder framework. A key contribution is the integration of a sparse motion dictionary, which is trained concurrently with the generative model to learn a disentangled representation of facial motion. Leveraging this motion dictionary, the paper introduces an "edit-warp-render" strategy to effectively address pose misalignment in cross-identity reenactment. Extensive experiments are presented to demonstrate the scalability of the proposed method.

**Strengths:**

1. **Novel Synthesis of Techniques:** The paper presents an elegant integration of two powerful and established methods: GAN and sparse dictionary learning, applying them effectively to the portrait animation domain.

2. **Demonstrated Scalability:** The experimental results successfully demonstrate the architecture's capacity to scale to high-resolution outputs and handle complex animation tasks.

**Weaknesses:**

**Insufficient Ablation of the Sparse Motion Dictionary:** The sparse motion dictionary is central to the paper's contributions, yet it is not sufficiently validated through detailed ablation studies. The specific benefits of the dictionary and its sparsity constraint are therefore not fully quantified.

**Questions:**

1. **Question about Sparse Motion Dictionary:**

    - **Architectural Details:** Could you please specify the scale and parameters of the motion dictionary, including the total number and the dimension of the embedding?

    - **Quantitative Sparsity:** Besides the visual comparsion on only one video sample, autho(s) should provide quantitative metrics that characterize the sparsity.

    - **Properties of the Motion Space:** The paper claims the motion embeddings are disentangled and interpretable. To substantiate this, could author(s) evaluate the interpolation properties of the learned motion space? For example, a comparison of the perceptual path length (PPL) in the motion space, with and without the sparsity constraint, would provide strong evidence for this claim.

2. **Question about Experiments:**

    - **User study:** Could author(s) conduct a user study to compare your method against baselines? This would provide a more robust measure of perceptual quality than automated metrics alone.

    - **Generalization:** Could author(s) provide qualitative examples of the model's performance on non-human characters (e.g., cartoons, avatars) to demonstrate its generalization capabilities?

3. **Question about Experiments Model Architecture and Design Rationale:**

    - **Architectural:** The paper mentions that the architecture is inspired by StyleGAN-T. For reproducibility, could author(s) provide more specific details about the encoder, generator, and discriminator architectures?

    - **Rationale for Model Selection:** Could author(s) offer more insight into the decision to use a GAN-based framework, especially given the well-documented training stability advantages of modern diffusion models? Furthermore, have author(s) considered whether the learned motion dictionary could be used to fine-tune a pre-trained video diffusion model to potentially achieve superior portrait animation performance?

---

### Official Review · Reviewer_sgtN · 2025-11-01

**Soundness:** 2
**Presentation:** 3
**Contribution:** 2
**Rating:** 4
**Confidence:** 4

**Summary:**

This paper presents LIA-X, an interpretable and controllable portrait animation framework that transfers facial dynamics from a driving video to a static source portrait. The method builds upon LIA by introducing a sparsity constraint on motion coefficients to encourage each motion basis vector in the dictionary to correspond to semantically meaningful and disentangled facial motions (e.g., yaw, pitch, mouth, eyes). Leveraging interpretable latent codes, the model supports fine-grained user-controlled editing before warping and rendering.

**Strengths:**

The Sparse Motion Dictionary introduces an interpretable regularization mechanism within the LIA framework. Although sparsity is a classical idea, applying it to disentangle latent motion vectors for human-readable semantics is a novel interpretability-driven adaptation. The edit-warp-render paradigm offers a conceptually clean way to integrate user control into a previously self-supervised latent animation pipeline.

**Weaknesses:**

1. The overall architecture (Encoder–Flow–Renderer) and loss formulation are largely inherited from LIA. The paper provides no theoretical justification or analytical evidence explaining why sparsity should induce semantic disentanglement in the motion dictionary. As a result, the contribution is primarily incremental.
2. The claimed “controllability” arises randomly from training statistics and is not guaranteed or reproducible across runs or identities.
3. The model lacks mechanisms (such as mutual information or attribute supervision) to consistently bind dictionary vectors to specific facial semantics. Sparsity alone cannot prevent cross-correlation between motion vectors and different vectors can still encode mixed semantics.
4. Excessive sparsity penalty could reduce motion fidelity, causing stiff or incomplete facial movements; this trade-off is not analyzed.
5. The paper does not discuss how the model performs under challenging cases such as extreme expressions, occlusions, or large head rotations, which are critical for assessing robustness.
6. Despite its interpretability improvements, LIA-X still inherits the visual quality of autoencoder-based methods—limited fidelity, over-smoothing, and inferior texture sharpness—compared to modern diffusion-based portrait animation approaches.
7. L1 regularization could introduce gradient instability in GAN training, but no visualization of training dynamics is given.

**Questions:**

please check the weakness part.

---

### Official Review · Reviewer_fJDV · 2025-11-01

**Soundness:** 2
**Presentation:** 3
**Contribution:** 2
**Rating:** 4
**Confidence:** 4

**Summary:**

This paper introduces LIA-X, a self-supervised autoencoder for controllable portrait animation that incorporates a Sparse Motion Dictionary to make motion codes interpretable. Unlike prior “warp-render” methods, LIA-X supports an “edit-warp-render” pipeline, enabling users to adjust head pose or facial expression before transferring motion from a driving video. The authors scale the model up to ~1B parameters, demonstrating improved performance and efficiency over GAN and diffusion baselines such as FOMM, LivePortrait, and LIA.

The key difference w.r.t the original LIA is the introduction of a sparsity constraint in the coefficients of motion codes, which results in "semantic" vectors. And the addition of scalability experiments with big networks.

**Strengths:**

Novelty & Interpretability – Introducing sparsity in the motion dictionary is a simple yet effective idea that yields human-interpretable motion vectors (e.g., yaw, pitch, smile).

Controllability – The edit-warp-render approach is well-motivated and addresses pose/expression misalignment issues common in talking-head animation.

**Weaknesses:**

Limited Theoretical Depth – The sparse dictionary idea, though effective, is conceptually straightforward (essentially regularization on motion coefficients). The paper lacks deeper analysis of why sparsity leads to disentanglement.

Evaluation Diversity – While results on self- and cross-reenactment are strong, the experiments are limited to face animation datasets; no user study or generalization to non-portrait domains is shown.

Scalability Ceiling – Gains saturate beyond 0.3B parameters (Tab. 3); the discussion attributes this to data limits but doesn’t provide strong evidence.

Writing – Though generally clear, some sections (especially 3.3 and 4.2) repeat similar ideas; could benefit from tighter exposition and clearer distinction between training and inference stages.

**Questions:**

NA

**Details Of Ethics Concerns:**

The ethics section is brief and does not discuss misuse risks (deepfakes) or possible mitigations, which are crucial for this topic.

---

### Note · Authors · 2025-11-14

I have read and agree with the venue's withdrawal policy on behalf of myself and my co-authors.